# Predictors of Intention to Vaccinate against COVID-19 in the General Public in Hong Kong: Findings from a Population-Based, Cross-Sectional Survey

**DOI:** 10.3390/vaccines9070696

**Published:** 2021-06-25

**Authors:** Elsie Yan, Daniel W. L. Lai, Vincent W. P. Lee

**Affiliations:** 1Department of Applied Social Sciences, The Hong Kong Polytechnic University, Hong Kong 999077, China; vwp.lee@polyu.edu.hk; 2Faculty of Social Sciences, Baptist University Hong Kong, Hong Kong 999077, China; daniel_lai@hkbu.edu.hk

**Keywords:** Hong Kong, COVID-19, intention to vaccinate, health belief model

## Abstract

Vaccination is one of the most effective ways to stop the spread of COVID-19. Understanding factors associated with intention to receive COVID-19 vaccines is the key to a successful vaccination programme. This cross-sectional study explored the rate of vaccination intention and identified its predictors using the health belief model (HBM) in the general population in Hong Kong during the pandemic. Data were collected between December 2020 and January 2021 via telephone surveys. Hierarchical logistic regression analysis was used to identify factors associated with intention to receive COVID-19 vaccines. A total of 1255 adults (>18 years, 53% female) completed the telephone survey. Overall, 42% indicated an intention to vaccinate, 31.5% showed vaccine hesitancy, and 26.5% reported refusal to receive any COVID-19 vaccines. Individuals who were men, older in age, working, with past experiences of other pandemics, less concerned with the vaccine safety, with poorer knowledge about COVID-19, and having greater levels of perceived susceptibility, self-efficacy, cues to action, and acceptance of governmental preventive measures related to COVID-19 were significantly more likely to report an intention to vaccinate. The low intention among the Hong Kong population reflects the importance of developing effective vaccination promotion campaigns with the predictors identified in this study.

## 1. Introduction

Since the first case was reported in China in December 2019, the Coronavirus Disease 2019 (COVID-19) has spread quickly worldwide, leading to tragic loss of many human lives and unimaginable burden on global health and economy. On 11 March 2020, the WHO declared the COVID-19 outbreak in China as a pandemic. The world’s hopes of resuming normal lives have been pinned on the development of COVID-19 vaccines. Along with disease preventive measures such as wearing of masks and social distancing, vaccination, which is believed to be one of the most effective ways of halting the spread of virus, has been central to the recovery strategy across the globe [1]. To reach a state of herd immunity, widespread vaccination is crucial. In the case of COVID-19, it has been estimated that at least 60% of the populations must develop immunity against the pathogen, in a way that COVID-19 transmission through human interactions can be notably reduced [2]. Since achieving population immunity through natural means may inflict unprecedented harm and danger, vaccination is thus the preferred option to combat COVID-19. As of 21 May 2021, there are 19 vaccines within the WHO evaluation process [3]. Clinical trials have revealed encouraging results, indicating the effectiveness of the existing vaccines [4,5]. However, alongside the development of the COVID-19 vaccines, there has been a growing mistrust in their safety globally [6], and such mistrust often results in hesitancy in vaccine uptake among the general public.

Vaccine hesitancy poses specific challenges to the success of vaccination programmes as it could greatly reduce the rate of vaccine uptake [7]. In 2019, it was listed by the WHO as one of the ten major threats to global health [8]. It has been shown in a review of 23 academic research studies and 103 opinion polls that COVID-19 vaccine hesitancy is increasing in the general public over time, leading to an alarming problem that hinders the achievement of herd immunity in the world [9]. Intention to receive the COVID-19 vaccines varies greatly across countries. A systematic review of vaccine acceptance found that the rates might vary from 24% (Kuwait) to 97% (Ecuador) [10]. In Hong Kong, one of the first cities being affected by the disease, researchers have consistently found low intentions to receive COVID-19 vaccines: Only 35–44% of the population would get vaccinated when the vaccines are available [11,12], a percentage far lower than that required to reach herd immunity.

To design an effective vaccination programme to maximise the vaccine uptake, a thorough understanding of the factors influencing individuals’ attitudes towards vaccination is of substantial importance. The World Health Organization (WHO) has highlighted the effects of individual confidence in vaccines, perceived values of the vaccines, and perceived obstacles to get access to the vaccines as three main factors contributing to intention to vaccinate [13], whilst others suggest the application of different cognitive health behaviour theories to account for such intention. The Health Belief Model (HBM) is one of the most widely used theoretically frameworks to explain health behaviours including flu vaccination [14], hepatitis B vaccination [15], and COVID-19 vaccination [11]. This study set out to examine the intention to receive COVID-19 vaccines in the general populations, and to identify predictors of individuals’ intention to vaccinate right before the launch of the free vaccination programme in the city in February 2021. In particular, we adopted the HBM in the prediction of intention to receive the vaccines and expanded the prediction model by adding various socio-demographic characteristics and COVID-19 specific factors that have been found related to vaccine uptake in the literature [16,17,18,19,20]. It was hypothesised that intention to receive COVID-19 vaccines would be predicted by the HBM factors (including perceived susceptibility, perceived severity, perceived benefits, perceived barriers, self-efficacy, and cues to action), acceptance of the governmental measures to prevent COVID-19, concern about COVID-19, experience of contracting COVID-19, past experience of other pandemics, knowledge about COVID-19, concern about vaccine safety and side effects, trust in the healthcare system and the government, as well as socio-demographic characteristics including gender, age, education attainment, and economic activity status.

## 2. Materials and Methods

### 2.1. Study Design and Settings

This study employed a population-based, cross-sectional survey design. We conducted a random telephone survey from December 2020 to January 2021 in Hong Kong, during which the city was hit by the fourth wave of the COVID-19 epidemic. All individuals who were Hong Kong residents, 18 years old or above, and able to communicate in Chinese (Cantonese or Mandarin) were eligible to participate. To minimise possible sampling errors, we selected telephone numbers using a multiphase procedure. First, we drew random numbers from a directory covering both landline and mobile numbers in Hong Kong. The drawn numbers were used as seed numbers to generate another set of telephone sample with the last digit “±one/two” method. We then filtered the two sets of numbers to avoid duplication, and mixed the remaining numbers in random order to produce the final set of telephone samples. A team of trained research assistants was responsible to contact eligible participants by phone calls from 9:00 a.m. to 9:00 p.m. during weekdays under the close supervision of our research team. Eligible individuals were invited to participate in the survey, which consisted of about 50 items and took approximately 20–30 min to complete. In the case of landline numbers, when there was more than one eligible individual, one of them would be selected using the last birthday selection method. Oral informed consent was sought before each interview began. All data were randomly collected using a computer-assisted telephone interview (CATI) system, which allowed real-time data capture and consolidation, in an anonymous way. For telephone numbers with no response in the first attempt, four more attempts in separate occasions were given before they were classified as “non-contact.”

### 2.2. Measures

#### 2.2.1. Intention to Vaccinate

We measured one’s intention to get the COVID-19 vaccination using a single item. Participants were asked to rate the item (“When COVID-19 vaccines are available, I will definitely get the vaccination”) against a 5-point Likert scale, from “1” (strongly disagree) to “5” (strongly agree). According to their responses, participants were classified into one of the three groups: (a) those who intended to vaccinate (giving response 4/5), (b) those who were undecided (giving response 3), and (c) those who were unwilling to vaccinate (giving response 1/2).

#### 2.2.2. The HBM Factors

In this study, HBM factors included perceived susceptibility, perceived severity, perceived benefits, perceived barriers, self-efficacy, cues to action, and acceptance of governmental measures related to COVID-19 prevention (e.g., group gathering restrictions and regulations on wearing of mask). Perceived susceptibility was measured with the item (“It is likely for me to contract COVID-19”), while perceived severity was assessed with (“COVID-19 is a serious disease”). Perceived benefits and barriers were measured with three items and six items, respectively. Sample items included (“I believe that it could help prevent the spread of COVID-19”) for perceived benefits, and (“I believe it is unnecessary and meaningless.”) for perceived barriers. Self-efficacy was assessed with the 10-item Generalized Self-Efficacy Scale [21] (e.g., “I can usually handle whatever comes my way.”); and cues to action were assessed with seven items (e.g., “My doctor has recommended me to do it.”). On the other hand, acceptance of governmental measures related to COVID-19 prevention was assessed with six items. Participants were asked to rate their acceptance levels of six different governmental measures including: (a) the regulations on the wearing of mask, (b) the restrictions on dine-in services among restaurants, (c) the restrictions on group gatherings, (d) compulsory testing of COVID-19 for specific groups, (e) compulsory quarantine of individuals arriving Hong Kong, and (f) compulsory quarantine of certain groups of citizens when in need.

All items were rated on a 5-point Likert scale, from “1” (strongly disagree) to “5” (strongly agree). Item scores were averaged for each scale of the HBM factors. Higher mean scores reflected higher levels of the specific factors, except for the scale of perceived barriers of which the item scores were reverse-coded. In that case, a higher score of the perceived barriers scale (reversed) indicated a lower level of barriers perceived by the participant. Internal consistencies of the HBM factors were good in this study. Cronbach’s alpha was 0.78 for perceived benefits, 0.88 for perceived barriers, 0.92 for self-efficacy, 0.83 for cues to action, and 0.85 for acceptance to governmental measures.

#### 2.2.3. COVID-19 Specific Factors

In this study, we also measured the following variables: (a) participants’ concern over COVID-19 in general with the item (“I am concerned about COVID-19.”); (b) concern about the safety of COVID-19 vaccines with the item (“I am concerned about the safety and side effects of COVID-19 vaccines.”); (c) current experience related to COVID-19 with three items (e.g., “My family member has contracted COVID-19.”); and (d) past experience of other pandemics with two items (e.g., “I have contracted SARS/H1N1 before.”).

To test participants’ knowledge about COVID-19, we employed the COVID-19 Knowledge Scale [22], a 12-item scale assessing one’s knowledge on the symptoms, transmission routes, and possible prevention methods of COVID-19. Sample questions included (symptoms: “The major symptoms include fever, fatigue, dry cough, and myalgia”), (transmission route: “COVID-19 can be spread through respiratory droplets.”), and (prevention methods: “Wearing of masks can help stop spreading of COVID-19.”) Participants were asked to indicate whether the item was correct or not, by choosing from “true”, “false”, and “not sure”. Participants received one mark for every correct answer. Total scores ranged from 0 to 12, with a higher score reflecting better knowledge. In this study, the COVID-19 Knowledge Scale achieved a Cronbach’s alpha of 0.65, reflecting a satisfactory scale reliability.

Trust in authorities were assessed with two items, including (“I can trust the government in handling the COVID-19 pandemic.”) and {“I can trust the doctors, nurses, and other healthcare professionals in controlling the spread of COVID-19.”) The items were rated on a 5-point scale, from “1” (strongly disagree) to “5” (strongly agree). Item scores were averaged. Higher scores indicated greater trust in the specific authority.

#### 2.2.4. Demographic Factors

Demographic background of the participants, which included gender, age, highest education attained, and economic activity status, were captured for further analysis.

### 2.3. Statistical Analysis

Descriptive analyses were performed for all study variables. Participants were first classified into three groups according to their intention to vaccinate, including those who intended to vaccinate, those who were undecided, and those who were unwilling to get vaccination. The latter two were then grouped together as “undecided/unwilling” for further analysis. The use of two groups instead of three in the vaccination intention when conducting statistical analysis was to highlight the potential difference between those who intended to receive COVID-19 vaccines and those who did not, and to identify factors that significantly predict one’s intention to vaccinate. To ensure the representativeness, the raw data in this study were weighted according to the latest sex-age distribution and education distribution in the Hong Kong population. Using chi-square tests and *t*-tests, demographic factors, HBM factors, and COVID-19 specific factors were compared between those who intended to vaccinate and those who were undecided or unwilling to vaccinate.

To investigate the factors associated with intention to get the COVID-19 vaccination, we performed a hierarchical logistic regression. The intention to vaccinate was used as the dependent variable (“0” for undecided/unwilling to vaccinate and “1” for intended to vaccinate). The HBM factors, COVID-19 specific factors, and demographic factors were used as independent variables which were divided into four blocks. Demographic factors were entered into block 1; HBM factors were entered into block 2; followed by COVID-19 specific factors (concern over COVID-19 in general, concern about the safety and side effects of vaccines, current experience of COVID-19, past experience of other pandemics, and knowledge about COVID-19) in block 3. Trust in the authorities was the sole variable being entered in block 4. All HBM factors and COVID-19 specific factors were transformed to standard scores before entering the regression model, to ensure all scales were in the same metric. Odds ratios were adjusted with other variables in the regression model. Multicollinearity was checked before conducting the regression.

All analyses were done using SPSS 26.0. *p*-values smaller than 0.05 were deemed statistically significant.

## 3. Results

### 3.1. Intention to Vaccinate and Demographic Characteristics of Participants

In this study, we sampled 6000 telephone numbers using the multiphase sampling procedure. A total of 3216 numbers were identified as invalid (e.g., fax numbers or numbers not in use). Of the remaining 2784 numbers, 1255 eligible participants completed the telephone survey, giving a response rate of 45.1%. Excluded cases were mostly non-contact (34.8%), followed by refusal to participate (19.4%) and language barrier (0.8%).

After weighting, about 42.0% of the participants indicated an intention to get the COVID-19 vaccination, whilst 26.5% reported a refusal or unwillingness to vaccinate. A total of 31.5% were not yet decided during the survey period, showing hesitancy to the COVID-19 vaccines.

Table 1 summarises the demographic characteristics of the participants. Overall, 53.0% were women, 41.3% were 55 years old or above, and 45.6% had completed upper secondary education. A majority (61.7%) were working persons. We compared the demographic characteristics between those who intended to receive a COVID-19 vaccine and those who were undecided or unwilling to vaccinate, and found significant between-group differences among the four demographic factors. Specifically, individuals who were men (χ^2^ = 8.46, *p* < 0.05), older-aged (χ^2^ = 84.98, *p* < 0.001), and with lower education attained (χ^2^ = 48.37, *p* < 0.001) were more likely to report an intention to vaccinate, whilst working persons (χ^2^ = 73.76, *p* < 0.001) were more likely to report vaccine hesitancy or refusal. Comparing the distributions of intention, hesitancy, and unwillingness to vaccinate by age group, we observed an increasing trend of intention, a decreasing trend of hesitancy, and a J-shaped increasing trend of unwillingness to receive COVID-19 vaccines with increased age (Figure 1).

### 3.2. The HBM Factors and COVID-19 Specific Factors

Table 2 presents the mean scores and standard deviations of the HBM factors and COVID-19 specific factors in this study. Significant between-group differences were found in all of the studied HBM factors but self-efficacy (*t* = −1.10, *p* > 0.05). When compared to those who were undecided or unwilling to vaccinate, participants who intended to get vaccination showed greater levels of perceived susceptibility of contracting COVID-19 (*t* = −8.20, *p* < 0.001), perceived severity of the disease (*t* = −6.90, *p* < 0.001), perceived benefits (*t* = −8.12, *p* < 0.001), cues to action (*t* = −9.90, *p* < 0.001), and acceptance to the governmental measures to prevent COVID-19 (*t* = −4.83, *p* < 0.001). On the other hand, participants who were undecided or unwilling to get vaccination reported a greater level of perceived barriers (reverse score: *t* = −3.06, *p* < 0.01).

Concerning the COVID-19 specific factors, this study revealed significant differences between the two groups of participants in all variables studied. Results demonstrated that those who intended to receive the vaccines showed greater concern about COVID-19 in general (*t* = −7.55, *p* < 0.001), more experiences related to COVID-19 (*t* = −7.35, *p* < 0.001) and other pandemics (*t* = −8.65, *p* < 0.001), and more trust in authorities (*t* = −19.49, *p* < 0.001) than those who did not. In contrast, those who were undecided or unwilling to receive COVID-19 vaccines reported greater concern about the safety and side effects of the vaccines (*t* = 5.82, *p* < 0.001) and better knowledge about the disease (*t* = 2.67, *p* < 0.01).

### 3.3. Factors Associated with Intention to Receive COVID-19 Vaccines

Results of the hierarchical logistic regression analysis are shown in Table 3. Our first model, which included only demographic factors, explained 11.1% of the variance in intention to receive COVID-19 vaccines. In this model, male gender (OR = 1.53, 95% CI 1.20–1.95) and older age (OR = 1.03, 95% CI 1.02–1.04) positively predicted the likelihood to indicate an intention to vaccinate. In contrast, an economic activity of working (as compared to not working for a living) predicted a decreased likelihood of acceptance to COVID-19 vaccines (OR = 0.71, 95% CI 0.54–0.94).

When the HBM factors were entered into the model, an additional 17.4% of explained variance was observed. In this model, all HBM factors significantly predicted the intention to receive COVID-19 vaccines, except for perceived severity of COVID-19 of which the association was not statistically significant. Intention to receive vaccines was positively predicted by higher levels of perceived susceptibility (OR = 3.01, 95% CI 2.09–4.34), perceived benefits (OR = 1.32, 95% CI 1.11–1.58), self-efficacy (OR = 1.20, 95% CI 1.02–1.40), cues to action (OR = 1.55, 95% CI 1.31–1.84), and acceptance to governmental measures to prevent COVID-19 (OR = 1.21, 95% CI 1.01–1.45); as well as a lower level of perceived barriers (reverse score: OR = 1.66, 95% CI 1.40–1.97).

The inclusion of the COVID-19 specific factors added 8.4% of explained variance to the regression model. In the third model, concern about COVID-19 in general (OR = 1.20, 95% CI 1.02–1.41), past experience of other pandemics (OR = 1.72, 95% CI 1.39–2.13), and concern about the safety and side effects of vaccines (OR = 0.61, 95% CI 0.53–0.70) were significant predictors of the intention to receive vaccines. However, current experience of COVID-19 and knowledge about the disease were not significantly associated with vaccination intention here.

The final model, with all factors including trust in authorities were added, explained 47.2% of the variance in intention to get COVID-19 vaccination. In this model, after the adjustment of all other independent variables, participants who reported trust in authorities were 2.9 times more likely to show intention to receive vaccines (OR = 2.88, 95% CI 2.38–3.50). Compared with previous models, perceived benefits, perceived barriers, and concern about COVID-19 in general became non-significant predictors of the intention to receive vaccines when all other variables including trust in authorities were adjusted for. In contrast, knowledge about COVID-19 became a significant negative predictor of the intention to vaccinate (OR = 0.78, 95% CI 0.66–0.93) in the final model. That is, better knowledge significantly predicted a lower likelihood for participants to indicate an intention to receive COVID-19 vaccines.

## 4. Discussion

This study provides a reliable update on the rate of intention of the general public to receive COVID-19 vaccines in Hong Kong, and describes comprehensively the factors predicting vaccination intention using a population-based sample of Chinese adults. Findings have shown that, right before the beginning of the COVID-19 vaccination programme in February 2021 in Hong Kong, less only two-fifths (42%) of the population intend to receive the vaccines. Although the rate of intention revealed in this study is comparable to previous findings in the same population (35–44%) [11,12,23], it is still far lower than the rate of 60% required to achieve herd immunity and halt the community spread of the disease [2]. 

Our findings provide another piece of evidence that shows the low intention to get vaccinated in Hong Kong when compared to other countries in the world. For instance, other research has shown high vaccination intention or acceptance in the U.S. (62–75%) [17,24,25], UK (64%) [26], Australia (86%) [27], France (77–78%) [28,29], Japan (62–66%) [30,31], Bangladesh (69%) [32], Israel (80%) [18], and India (89%) [33]. Even when we compared the rates within the Chinese population, intention to receive COVID-19 vaccines in Hong Kong is among the bottom. Previous studies have found that 89–92% of individuals in China were willing to get vaccinated once the vaccines are available [34,35]. Our findings suggest possible discrepancies within a large population of the same ethnicity, and that low vaccination intention may not be an issue solely due to cultural reasons.

The low intention in the Hong Kong population has been relatively consistent across time. In a repeated measures survey study in Hong Kong [12], the rates of willingness to accept the COVID-19 vaccines were 44% and 35% when the city was facing the first wave (February 2020) and third wave of the epidemic (August 2020), respectively. The authors suggest a decline of the intention to receive COVID-19 vaccines over time, and their view has later been echoed by other researchers who have observed a slight drop in the acceptance of vaccination from 92% to 89% from March to December 2020 in China [35] and a large reduction from 74% to 50% from April to December in the U.S. [36]. However, our findings show that the trend of intention to receive a vaccine may not be a simple declining curve as time progresses. Instead, situations could be complicated and the vaccination intention among the general public could be fluctuating across different waves of the epidemic. Nonetheless, the time-robust low intention to receive COVID-19 vaccines among the general public in Hong Kong (35–44% in previous research [12,23] and 42% in this study) raises an alarming issue to the public health sector, and warrants the investigation on the ways to increase the acceptance and willingness to get vaccinated by identifying their potential predictors.

Congruent with previous findings, socio-demographic background has been demonstrated to be significant predictors of the intention to receive vaccines in this study. Our findings demonstrate that men are more likely than women to accept the vaccines, which is consistent with the majority of the literature [16,18,19]. The lower intention among women may be explained by some female-specific concerns over the vaccines. For example, a study in Jordan and Kuwait has revealed common false beliefs among the general public that COVID-19 vaccines may cause infertility [37]. As women, in general, tend to express greater worry and fear of the side effects of the vaccines [19], a greater vaccine hesitancy and refusal among them could be expected. Older age is another significant demographic factor that has been consistently demonstrated as a predictor to higher intention to receive COVID-19 vaccines [28,32]. Age may affect intention to vaccinate in several ways. On one hand, older individuals may be more willing to receive the vaccine as a means of protection since they are commonly believed as having high risks of contracting COVID-19 [18]; on the other hand, younger individuals may be more exposed to vaccine-related misinformation via social media [19], leading to greater vaccine hesitancy among the young populations [38].

Our findings show that the HBM could help explain intention to receive COVID-19 vaccines. According to the model, the likelihood of individuals to engage in health promoting behaviours (in this case, vaccine uptake) can be determined partly by their perceptions of the health threat and the related health behaviours. In this study, perceived susceptibility, one of the key HBM factors, is demonstrated as a significant predictor of vaccination intention. Individuals who perceive themselves as more prone to contracting COVID-19 tend to show higher intention to receive a vaccine. The association between perceived susceptibility and vaccination intention has been well documented in the literature [25,28,39]. For example, a study in Italy has shown that high self-perceived vulnerability to COVID-19 may lead to more acquiescent behaviours and greater perceived responsibility towards the community in the acceptance of vaccines [40]. In contrast, perceived severity, another key factor of the HBM, does not seem to be related to individuals’ intention to vaccinate. This finding contributes to the mixed evidence of the association between perceived severity and vaccination intention in the literature [11,16,24]. Whether individuals who perceive COVID-19 as a more severe disease would be more willing to receive the vaccine is still inconclusive at this stage. However, our findings do provide some insights into the public health sector in promoting vaccine uptake: Perceived susceptibility or risk of contracting COVID-19 could be effectively translated into health behaviours in the prevention of the disease. Future promotional messages may frame a high risk of the disease in influencing the general public’s intention to get vaccinated.

The significant predictive power of self-efficacy and cues to action, two other components of the HBM, revealed in this study also underscore the importance of setting up vaccine uptake promotion campaigns to deal with individuals with low intention. In line with previous research [18,32], this study finds that individuals with greater self-efficacy in handling difficulties in life and those exposed to more cues to the vaccine uptake tend to express higher intention to get vaccinated. Future vaccine-related campaigns may promote general self-efficacy and cover more cues to action. For example, more resources can be invested in information campaigns that broadcast health professionals’ advice or encourage individuals to share their positive experiences regarding vaccination with their friends and relatives through social media to help build confidence and subjective norms in the general public.

Interestingly, perceived benefits and perceived barriers did not play a significant role in predicting the intention to receive COVID-19 vaccines after the adjustment for other variables including trust in the authorities. In this study, when trust in authorities was added to the prediction model, perceived benefits, perceived barriers, and concern about COVID-19 became non-significant predictors. It has been suggested that worry and concern about COVID-19 in general could alter one’s beliefs [41], which may include those on the potential benefits and obstacles related to a vaccine: Individuals who show higher levels of worry and concern are more likely to perceive vaccines as great and effective, while at the same considering barriers to receive the vaccines too hard to overcome. In this sense, perceived benefits, perceived barriers, and concern about COVID-19 may be inter-correlated. Our findings show that the associations of these three factors with vaccination intention could be “cancelled out” when trust in authorities is present, implying the potential power of trust in predicting intention to receive the vaccines. Public trust in the healthcare professionals and the government as well as individuals’ acceptance of governmental preventive measures related to COVID-19 have stood out as a strong predictor of acceptance of vaccination in both literature and this study [42,43]. Individuals having greater faith in the healthcare system and the government may show greater confidence that the spread of COVID-19 can be stopped in the community. They may be less worried and concerned about the disease in general, and more willing to receive vaccinations when perceived benefits and barriers are not altered.

Apart from the demographic and HBM factors, we also identified several COVID-19 specific predictors in this study. Parallel to other studies [20,24,27,34], this study reveals a negative association between concern about vaccine safety and intention to receive the vaccine. That is, the more an individual is concerned about the safety issues, the less likely he or she will receive the vaccine. As of January 2021, when the survey study was conducted, most COVID-19 vaccines were still in early phases of the clinical trials. The use of new technologies such as mRNA technology in vaccine development, the relatively short duration of clinical trial period, the lack of large-scale research study on the effectiveness, and the unclear long-term side effects of the COVID-19 vaccines may all lead to great concern and worry about their safety issues among the general public, which in turn reduce individuals’ motivation to get vaccinated at an early stage. 

It may be particularly noteworthy that knowledge is a negative predictor of intention to receive the COVID-19 vaccines, which provides some divergent evidence for the positive association between the two as found in past research [17,43]. It has been widely believed that vaccine hesitancy is associated with misinformation and poor knowledge [19,44]. However, this study shows that individuals with better knowledge about COVID-19 can be less likely to indicate vaccination intention. This finding may suggest that, instead of a simple negative association, there can be a more complicated relationship between knowledge or information and intention to receive vaccines. However, our study assessed the general knowledge about COVID-19 but not the specific knowledge about COVID-19 vaccines. Whether the differences in these two types of knowledge matter is unknown. Future research may explore the association between knowledge and vaccination intention by assessing both general knowledge about COVID-19 and specific knowledge about its vaccines at the same time.

Novel to this study is that we also examined individuals’ experience of other pandemics and its association with intention to receive COVID-19 vaccines. Past experiences of vaccination have often been shown to be predictive of current vaccination intention [40]. However, whether past experiences of other pandemics (e.g., Severe Acute Respiratory Syndrome or SARS, Swine Flu or H1N1) in general could predict intention to receive COVID-19 vaccines was unclear. Our finding extends current knowledge by demonstrating that experiences of other pandemics could result in higher vaccination intention. This sheds light on future vaccination promotion, in which past experiences of other pandemics may be emphasised among certain groups of individuals to increase their likelihood of receiving vaccination.

To boost COVID-19 vaccine uptake, governments all over the world have been implementing various initiatives to encourage their citizens to take the action. Most initiatives focus on highlighting benefits, reducing barriers, and promoting cues to vaccinate. Some countries have introduced paid vaccination leave, with which employees are granted paid leave of several hours to a day after each dose of vaccination, [45,46], whilst others have proposed or launched the “vaccine passport” (a.k.a. “Green Pass”, “COVID Certificate”, “Coronapas”, “VaccineGuard” and “Excelsior Pass”) schemes [47,48,49]. Under these schemes, individuals with immunity, whether based on a past recovery from COVID-19 or a completion of two doses of recognised vaccines, may be exempted from free movement restrictions or quarantine policy when traveling, and granted free access to social, sports, and cultural events. By allowing immunised individuals to move freely and progressively return to normal life, vaccine passport schemes may speed up the vaccination uptake in the society. However, there have been doubts on relevant ethical and safety issues. For example, those who are not able to receive vaccines due to underlying health conditions will be automatically excluded from the schemes. Unless further evidence is available for the safety and effectiveness of alternative therapies or treatments to help these individuals develop immunity (e.g., monoclonal antibody therapy) [50], it could be unethical to “penalise” them for being unfit to vaccinate. As to safety issues, there have been concerns driven by uncertainties about the length of time that vaccine-built protection could maintain. Evidence is not yet extensive for how far existing vaccines could prevent COVID-19 transmission, and how high the risk of contracting COVID-19 would be after receiving two doses of vaccines is still unknown. Whether it is safe and secure for a group of vaccinated individuals to gather without proper disease preventive measures (such as mask wearing and social distancing) requires further research.

Like other research, there are several limitations to this study. First, our sample was not truly randomised, although we have employed a multiphase sampling procedure in drawing telephone numbers and weighted the data according to the sex-age and education attainment distribution of the Hong Kong population. Second, our study was cross-sectional, which limited the inferences about temporal precedence of the relationships among studied variables. Longitudinal studies that track individuals’ vaccination intention and actual vaccine uptake behaviour across time may help identify the temporal relationships of the variables. The third limitation of this study is the use of self-reported data. Biases and errors may appear when participants recalled their memory. Furthermore, this study explored vaccination intention in the general public only, and findings may not be generalised to specific groups of individuals. One example would be healthcare workers, who can be at high risk of exposure to the coronavirus. Previous research has explored the vaccination intention among healthcare workers worldwide, and findings indicated a huge gap across countries: For example, about 63% of nurses were willing to receive COVID-19 vaccines in Hong Kong [51], whilst only 23% of healthcare workers indicated an intention to vaccinate in Taiwan [52]. Whether healthcare workers are more or less willing than the general population to receive the vaccines is unclear. Another group that is worth investigating are frail or immunosuppressed individuals (e.g., elderly people with chronic health conditions or cancer patients receiving immunosuppressive therapy). Although most of them are considered safe to receive COVID-19 vaccines, a lower willingness to vaccinate could be expected, especially when the safety and efficacy of COVID-19 vaccines are yet fully known among those individuals at this stage. Clearly, further research on the intention among specific groups of individuals is necessary. A better understanding of their intention will certainly help develop specific vaccination promotion programmes that are effective for different groups. At last, the exclusion of non-Chinese speaking population (0.8%) may reduce the generalisability of the study findings. Future studies should include all residents in Hong Kong to minimise any bias due to ethnic and cultural differences.

## 5. Conclusions

The success of COVID-19 vaccines depends greatly on the high rate of vaccination uptake in the population. Our findings suggest a relatively low intention to receive COVID-19 vaccines right before the beginning of the free vaccination programme in Hong Kong, and shows that the low intention is time-robust when compared with previous studies in the same population [12]. This study identifies several predictors of individuals’ intention to vaccinate, and sheds light on future campaigns to promote vaccine uptake. It is recommended that future campaigns and programmes should consider individuals’ attitudes and perceptions towards COVID-19 and its vaccines, and build public confidence in the vaccination by emphasising the risks of the disease, promoting trust in the healthcare system and the government, providing more cues to action, and giving information to reduce the worry about the vaccines. To speed up the public vaccine uptake so as to achieve herd immunity as early as possible, effective promotion and communication to the general public should begin now.

## Figures and Tables

**Figure 1 vaccines-09-00696-f001:**
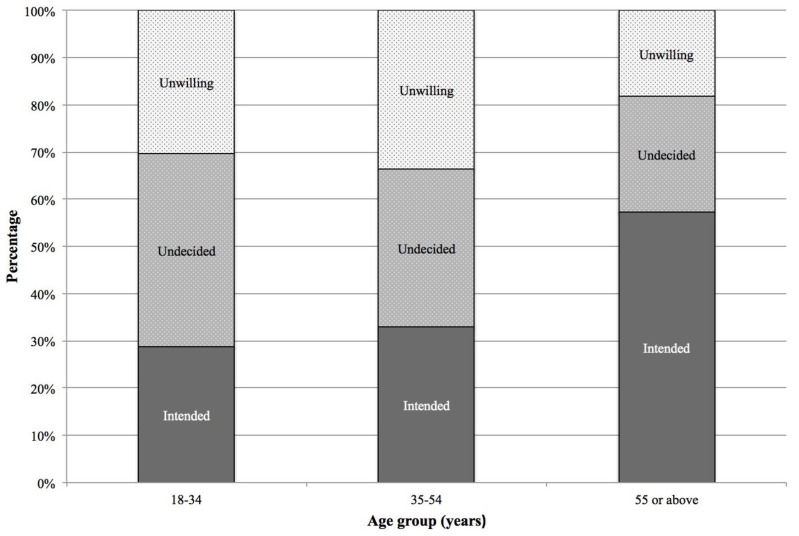
Distribution of intention to receive COVID-19 vaccines, by age group.

**Table 1 vaccines-09-00696-t001:** Distribution of demographic characteristics of the sample (Unweighted *n* = 1255).

Variable	Total			Intention to Vaccinate						Chi-Square	*p*-Value
				Intended (42.0%)			Undecided or Unwilling (58.0%)				
	*n*	Weighted *n*	%	*n*	Weighted *n*	%	*n*	Weighted *n*	%		
**Gender**										8.46	0.004
Female	657	3,252,280	53.0	251	1,252,453	48.6	406	1,999,827	56.3		
Male	598	2,881,760	47.0	277	1,327,178	51.4	321	1,554,582	43.7		
**Age (years)**										84.98	<0.001
18–34	294	1,453,440	23.7	84	417,491	16.2	210	1,035,949	29.1		
35–54	442	2,147,800	35.0	147	709,726	27.5	295	1,438,074	40.5		
55 or above	519	2,532,800	41.3	297	1,452,414	56.3	222	1,080,386	30.4		
**Education attainment**										48.37	<0.001
Primary/Lower secondary	392	1,927,535	31.4	216	1,063,473	41.2	176	864,062	24.3		
Upper secondary	575	2,795,408	45.6	228	1,106,664	42.9	347	1,688,745	47.5		
Diploma/University degree/Postgraduate degree	288	1,411,096	23.0	84	409,493	15.9	204	1,001,603	28.2		
**Economic activity status**										73.76	<0.001
Working person	775	3,782,101	61.7	278	1,352,301	52.4	497	2,429,799	68.4		
Student	62	302,632	4.9	16	78,273	3.0	46	224,359	6.3		
Homemaker	143	695,093	11.3	61	298,187	11.6	82	396,907	11.2		
Retired	223	1,096,609	17.9	148	727,142	28.2	75	369,467	10.4		
Unemployed/Between jobs	50	247,792	4.0	24	118,748	4.6	26	129,044	3.6		
Others	2	9813	0.2	1	4979	0.2	1	4834	0.1		

Note. Data were weighted according to the latest sex-age distribution and education attainment distribution in the Hong Kong population.

**Table 2 vaccines-09-00696-t002:** Mean scores and standard deviations of the variables, by participants’ intention to vaccinate.

Variable	Total		Intention to Vaccinate				*t*	*p*-Value
			Intended (42%)		Undecided/Unwilling (58%)			
	Mean	SD	Mean	SD	Mean	SD		
**Health Belief Model factors**								
Perceived susceptibility	2.19	0.39	2.30	0.46	2.11	0.32	−8.20	<0.001
Perceived severity	4.53	0.68	4.67	0.60	4.42	0.72	−6.90	<0.001
Perceived benefits	4.23	0.67	4.41	0.62	4.11	0.67	−8.12	<0.001
Perceived barriers (reversed)	2.54	1.00	2.64	1.25	2.46	0.77	−3.06	0.002
Self efficacy	3.41	0.73	3.44	0.90	3.39	0.57	−1.10	0.271
Cues to action	4.07	0.58	4.26	0.58	3.94	0.54	−9.99	<0.001
Acceptance of governmental measures related to COVID-19	4.40	0.66	4.50	0.57	4.33	0.72	−4.83	<0.001
**COVID-19 specific factors**								
Concern about COVID-19	4.07	0.93	4.29	0.85	3.90	0.95	−7.55	<0.001
Experience of COVID-19	2.20	1.12	2.47	1.29	2.00	0.93	−7.35	<0.001
Past experience of other pandemics	1.66	1.14	1.99	1.40	1.41	0.82	−8.65	<0.001
Concern about the safety of COVID-19 vaccines	3.88	0.95	3.69	0.98	4.01	0.90	5.82	<0.001
Knowledge about COVID-19	9.32	2.34	9.11	2.32	9.48	2.34	2.67	0.008
Trust in authorities	3.54	0.88	4.02	0.68	3.19	0.84	−19.49	<0.001

**Table 3 vaccines-09-00696-t003:** Hierarchical regression model of factors associated with intention to receive COVID-19 vaccines.

Variable	Intention to Vaccinate			
	Model 1	Model 2	Model 3	Model 4
	OR (95% CI)	OR (95% CI)	OR (95% CI)	OR (95% CI)
**Demographic factors**				
Gender (ref. Female)				
Male	1.53 *** (1.20–1.95)	1.46 ** (1.12–1.90)	1.48 ** (1.12–1.95)	1.53 ** (1.14–2.07)
Age	1.03 *** (1.02–1.04)	1.02 *** (1.01–1.03)	1.02 ** (1.01–1.03)	1.02 ** (1.00–1.03)
Education (ref. Diploma/University degree/Postgraduate degree)				
Primary/Lower secondary	1.29 (0.85–1.96)	1.66 * (1.05–2.62)	1.38 (0.85–2.23)	1.25 (0.75–2.10)
Upper secondary	1.19 (0.86–1.65)	1.35 (0.94–1.93)	1.37 (0.94–1.99)	1.24 (0.83–1.86)
Economic activity (Non-working)				
Working	0.71 * (0.54–0.94)	0.65 ** (0.48–0.87)	0.63 ** (0.46–0.86)	0.63 ** (0.45–0.87)
**Health Belief Model factors**				
Perceived susceptibility (ref. Not susceptible)		3.01 *** (2.09–4.34)	2.85 *** (1.94–4.19)	2.41 *** (1.60–3.64)
Perceived severity		1.09 (0.94–1.26)	0.98 (0.83–1.15)	1.03 (0.86–1.22)
Perceived benefits		1.32 ** (1.11–1.58)	1.35 ** (1.12–1.63)	1.18 (0.97–1.44)
Perceived barriers (reversed)		1.66 *** (1.40–1.97)	1.38 ** (1.14–1.68)	1.18 (0.96–1.46)
Self efficacy		1.20 * (1.02–1.40)	1.15 (0.97–1.36)	1.22 * (1.02–1.47)
Cues to action		1.55 *** (1.31–1.84)	1.64 *** (1.36–1.96)	1.31 ** (1.07–1.60)
Acceptance of governmental measures related to COVID-19		1.21 * (1.01–1.45)	1.32 ** (1.08–1.60)	1.24 * (1.01–1.53)
**COVID-19 specific factors**				
Concern about COVID-19			1.20 * (1.02–1.41)	1.13 (0.95–1.34)
Experience of COVID-19			0.93 (0.76–1.13)	1.05 (0.85–1.31)
Past experience of other pandemics			1.72 *** (1.39–2.13)	1.46 *** (1.17–1.83)
Concern about the safety of COVID-19 vaccines			0.61 *** (0.53–0.70)	0.61 *** (0.52–0.71)
Knowledge about COVID-19			0.88 (0.75–1.03)	0.78 ** (0.66–0.93)
Trust in authorities				2.88 *** (2.38–3.50)
**Model statistics**				
Cox and Snell *R*^2^	0.082	0.212	0.274	0.351
Nagelkerke *R*	0.111	0.285	0.369	0.472

Note. * *p* < 0.05. ** *p* < 0.01. *** *p* < 0.001.

## Data Availability

Materials and anonymous data are available from the authors by request.

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
