# Peer review of "Predictors of Intention to Vaccinate against COVID-19 in the General Public in Hong Kong: Findings from a Population-Based, Cross-Sectional Survey"

_vaccines, 2021, doi:10.3390/vaccines9070696_

Round 1

Reviewer 1 Report

It is an absolute relevant topic and the study is well conducted and described. I have only some minor points I would like to see addressed:

  • The HBM-factors are measured on 5-point Likert scale, whereas the COVID-19 specific factors are measured in mixed metric, thus, entering all predictors in one model, in requires the same metric, therefore the authors are urged to z-standardize their predictors
  • Logistic regression was performed, however nominal regression with 3 groups according to 2.2.1. Intention to vaccinate …is expected by the reader, since the authors stated: According to their responses, participants were classified into one of the three groups: (a) those who intended to vaccinate (giving response 4/5), (b) those who were undecided (giving response 3), and (c) those who were unwilling to vaccinate (giving response 1/2). At least, provide a rationale why do you omit category b
  • Please report reliability of the measures
  • Please check normality assumption of the HBM-factors

Author Response

Comment 1:

The HBM-factors are measured on 5-point Likert scale, whereas the COVID-19 specific factors are measured in mixed metric, thus, entering all predictors in one model, in requires the same metric, therefore the authors are urged to z-standardize their predictors.

Response:

We agree with this comment, and would like to thank the reviewer for pointing this out. We have conducted the regression analysis with all HBM factors and COVID-19 specific factors transformed into z-scores. We have revised all parts related:

(p.4) “All HBM factors and COVID-19 specific factors were transformed to standard scores before entering the regression model, to ensure all scales were in the same metric.”

(p.9) “Intention to receive vaccines was positively predicted by higher levels of perceived susceptibility (OR = 3.01, 95% CI 2.09-4.34), perceived benefits (OR = 1.32, 95% CI 1.11-1.58), self-efficacy (OR = 1.20, 95% CI 1.02-1.40), cues to action (OR = 1.55, 95% CI 1.31-1.84), and acceptance to governmental measure to prevent COVID-19 (OR = 1.21, 95% CI 1.01-1.45); as well as a lower level of perceived barriers (reverse score: OR = 1.66, 95% CI 1.40-1.97).”

(p.9) “The inclusion of the COVID-19 specific factors added 8.4% of explained variance to the regression model. In the third model, concern about COVID-19 in general (OR = 1.20, 95% CI 1.02-1.41), past experience of other pandemics (OR = 1.72, 95% CI 1.39-2.13), and concern about the safety and side effects of vaccines (OR = 0.61, 95% CI 0.53-0.70) were significant predictors of the intention to receive vaccines. Yet, current experience of COVID-19 and knowledge about the disease were not significantly associated with vaccination intention here.”

(p.9-p.10) Table 3

(p.10) “The final model, with all factors including trust in authorities were added, explained 47.2% of the variance in intention to get COVID-19 vaccination. In this model, after the adjustment of all other independent variables, participants who reported trust in authorities were 2.9 times more likely to show intention to receive vaccines (OR = 2.88, 95% CI 2.38-3.50). Compared with previous models, perceived benefits, perceived barriers, and concern about COVID-19 in general became non-significant predictors of the intention to receive vaccines when all other variables including trust in authorities were adjusted for. In contrast, knowledge about COVID-19 became a significant negative predictor of the intention to vaccinate (OR = 0.78, 95% CI 0.66-0.93) in the final model. That is, better knowledge significantly predicted a lower likelihood for participants to indicate an intention to receive COVID-19 vaccines.”

Comment 2:

Logistic regression was performed, however nominal regression with 3 groups according to 2.2.1. Intention to vaccinate …is expected by the reader, since the authors stated: According to their responses, participants were classified into one of the three groups: (a) those who intended to vaccinate (giving response 4/5), (b) those who were undecided (giving response 3), and (c) those who were unwilling to vaccinate (giving response 1/2). At least, provide a rationale why do you omit category b.

Response:

The reviewer has raised an important point here. First we would like to apologise if the manuscript was not written clearly enough. Group (b) participants were not omitted in the analysis; instead, they were grouped together with group (c) participants to form the category “undecided/unwilling” in the comparisons and logistic regression model.

We agree that it would have been interesting to compare the three groups (intended/undecided/unwilling to receive the vaccine) and explore the predictors for each of these decisions. However, in the case of this study, it seems more appropriate to perform two-group comparisons and logistic regression because the main objective of this study was to identify significant factors predicting an intention to vaccine (as compared to hesitancy/refusal), and to give insights on how we could promote vaccination in the studied population.

The combination of participants who indicated hesitancy and refusal in regression analysis has also been adopted in some existing studies, e.g. Wong et al. (2021).

To explain the rationale for the use of two groups instead of three in the analysis, We have added the following statement in the revised manuscript:

(p.4) “The use of two groups instead of three in the vaccination intention when conducting statistical analysis was to highlight the potential difference between those who intended to receive COVID-19 vaccines and those who did not, and to identify factors that significantly predict one’s intention to vaccinate.”

Reference:

Wong, M.C.S.; Wong, E.L.Y.; Huang, J.; Cheung, A.W.L.; Law, K.; Chong, M.K.C.; Ng, R.W.Y.; et al. Acceptance of the COVID-19 vaccine based on the health belief model: A population-based survey in Hong Kong. Vaccine 2021, 39, 1148-1156.

Comment 3:

Please report reliability of the measures.

Response:

Thanks to the reviewer for pointing this out. We have included the reliability in the revised manuscript accordingly:

(p.3) “Internal consistencies of the HBM factors were good in this study. Cronbach’s alpha was 0.78 for perceived benefits, 0.88 for perceived barriers, 0.92 for self-efficacy, 0.83 for cues to action, and 0.85 for acceptance to governmental measures.”

(p.4) “In this study, the COVID-19 Knowledge Scale achieved a Cronbach’s alpha of 0.65, reflecting a satisfactory scale reliability.”

Comment 4:

Please check normality assumption of the HBM-factors.

Response:

Thanks for this suggestion. Normality assumption is an important issue in most regression analyses. Yet, this study used logistic regression, which does not require residuals to follow a normal distribution. Therefore, we have not included the results of normality testing in the manuscript.

Reviewer 2 Report

Dear Editor

 The paper by YAN, Chau Wai Elsie and coworkers entitled:” Predictors of intention to vaccinate against COVID-19 in the 2 general public in Hong Kong: Findings from a population- 3 based, cross-sectional survey”

The title reflects the main subject of the study.

The abstract summarize and reflect the work described.

I have a minor comments:

I think this article can further improve by adding in discussion/ conclusion  section:

- In non-vaccine-COVID-19 respondents can passive immunization with MoAbs be considered?

- What strategy to adopt in those who have SARS-CoV-2 infection after the first or second dose of vaccine?

-How do I evaluate the COVID-19 vaccination response in frail and immunodepressed people?

-How can the COVID-19 vaccination response be evaluated in people exposed to the virus (HCW)?

-Can recalls be foreseen with vaccines other than the one used the first time?

According to what criteria to consider an immunized subject/ not at risk of transmitting the infection (towards the green pass)?.

Is quarantine still necessary in vaccinated people?

Author Response

Comment 1:

I think this article can further improve by adding in discussion/ conclusion section: 

In non-vaccine-COVID-19 respondents can passive immunization with MoAbs be considered?

Response:

We would like to thank the reviewer for the insightful suggestions. Monoclonal antibody therapy is a novel treatment for COVID-19 that just received emergency use authorization (EUA) by the U.S. Food and Drug Administration recently. We believe that it could be a possible alternative for individuals who are not suitable for vaccination to build immunity, but further evidence is required for its safety and effectiveness in the long-term. We have revised our manuscript to include a brief discussion on it:

(p.13) “To boost COVID-19 vaccine uptake, governments all over the world have been implementing various initiatives… However, there have been doubts on relevant ethical and safety issues. For example, those who are not able to receive vaccines due to underlying health conditions will be automatically excluded from the schemes. Unless further evidence is available for the safety and effectiveness of alternative therapies or treatments to help those individuals develop immunity (e.g. monoclonal antibody therapy) [50], it could be unethical to “penalise” them for being unfit to vaccinate...”

Comment 2:

What strategy to adopt in those who have SARS-CoV-2 infection after the first or second dose of vaccine?

Response:

The reviewer has raised an important issue about the vaccine protection and its relevant problems. We have also noticed that existing vaccines are not perfect to protect all vaccinated individuals, and there is a lack of longitudinal evidence to show the protection rates over time. It would have been interesting to have a detailed discussion on this issue and give suggestions on strategies to help those who are not protected after the completion of two doses of vaccines. Unfortunately, our data did not include relevant information, and our current findings do not provide any support for an empirically based discussion on this matter.

Yet, we agree with the reviewer that this could be a point worthy of investigation. We have incorporated it into our discussion in the revised manuscript:

(p.13) “… As to safety issues, there have been concerns driven by uncertainties about the length of time that vaccine-built protection could maintain. Evidence is not yet extensive for how far existing vaccines could prevent COVID-19 transmission, and how high the risk of contracting COVID-19 would be after receiving two doses of vaccines is still unknown. Whether it is safe and secure for a group of vaccinated individuals to gather without proper disease preventive measures (such as mask wearing and social distancing) requires further research.”

Comment 3:

How do I evaluate the COVID-19 vaccination response in frail and immunodepressed people?

Response:

We believe that the reviewer has made a very good point here. Vaccination intention and response among frail and immunodepressed people (e.g. older people with chronic health conditions and cancer patients receiving immunosuppressive therapy) is an issue that can never be overlooked. These individuals may represent those who have lower intention to receive vaccines, especially when safety of the COVID-19 vaccines among frail individuals has not been widely supported in some vaccines and efficacy of vaccines in immunodepressed or immunosuppressed individuals are not fully known. But in fact, most of these individuals are believed to be safe for receiving the COVID-19 vaccines in our sample. It is of great importance to evaluate their intention in order to develop effective vaccination promotion programmes for these specific groups of people.

We have strengthened the discussion by highlighting the need for future studies to explore the vaccination response among frail and immunodepressed individuals:

(p.14) “Further, this study explored vaccination intention in the general public only, and findings may not be generalized to specific groups of individuals... Another group that worth investigating are frail or immunosuppressed individuals (e.g. elderly people with chronic health conditions or cancer patients receiving immunosuppressive therapy). Although most of them are considered safe to receive COVID-19 vaccines, a lower willingness to vaccinate could be expected, especially when the safety and efficacy of COVID-19 vaccines are yet fully known among those individuals at this stage. Clearly, further research on the intention among specific groups of individuals is necessary. A better understanding of their intention will certainly help develop specific vaccination promotion programmes that are effective for different groups.

Comment 4:

How can the COVID-19 vaccination response be evaluated in people exposed to the virus (HCW)?

Response:

We agree with the reviewer that evaluating the vaccination intention among healthcare workers and other specific groups of individuals can be both interesting and meaningful. Unfortunately, we aimed at focusing on the general public in this study and did not include any investigation among healthcare workers. Yet, we have incorporated some discussion in the discussion section to highlight the importance of further research on this issue:

(p.14) “Further, this study explored vaccination intention in the general public only, and findings may not be generalized to specific groups of individuals. One example would be healthcare workers, who can be at high risk of exposure to the coronavirus. Previous research has explored the vaccination intention among healthcare workers worldwide, and findings indicated a huge gap across countries: For example, about 63% of nurses were willing to receive COVID-19 vaccines in Hong Kong [45], whilst only 23% of healthcare workers indicated an intention to vaccinate in Taiwan [46]. Whether healthcare workers are more or less willing than the general population to receive the vaccines are unclear... Clearly, further research on the intention among specific groups of individuals is necessary. A better understanding of their intention will certainly help develop specific vaccination promotion programmes that are effective for different groups.”

Comment 5:

Can recalls be foreseen with vaccines other than the one used the first time?

Response:

Here, the reviewer has questioned whether vaccine recalls could be expected. This is indeed an interesting question, but one that we believe lies beyond the scope of this study. We did not put intend to cover this aspect when we designed this study. Our current dataset and findings do not give any empirical support for us to comment on the likelihood of such recalls. We too look forward to a future where the field better understands the effectiveness of the existing vaccines, and where researchers are in a position to use the knowledge to determine whether recall of any vaccine is required.

Comment 6:

According to what criteria to consider an immunized subject/ not at risk of transmitting the infection (towards the green pass)?

Response:

We thank the reviewer for raising this question. We have included a brief discussion in the revised manuscript:

(p.13) “To boost COVID-19 vaccine uptake, governments all over the world have been implementing various initiatives to encourage their citizens to take the action. Most initiatives focus on highlighting benefits, reducing barriers, and promoting cues to vaccinate. Some countries have introduced paid vaccination leave, with which employees are granted paid leave of several hours to a day after each dose of vaccination, [45, 46]; whilst others have proposed or launched the “vaccine passport” (a.k.a. “Green Pass”, “COVID Certificate”, “Coronapas”, “VaccineGuard” and “Excelsior Pass”) schemes [47, 48, 49]. Under these schemes, individuals with immunity, whether based on a past recovery from COVID-19 or a completion of two doses of recognised vaccines, may be exempted from free movement restrictions or quarantine policy when traveling, and granted free access to social, sports, and cultural events. By allowing immunised individuals to move freely and progressively return to normal life, vaccine passport schemes may speed up the vaccination uptake in the society…

Comment 7:

Is quarantine still necessary in vaccinated people?

Response:

Another interesting question that we are looking forward to an answer in the future. As of today (19 June, 2021), there has not yet been a definite conclusion on whether vaccinated individuals should be exempted from quarantine restrictions. Some believe protection of vaccines is strong enough to allow vaccinated people to move freely, while other are in doubt. We believe that there is an urgent need for longitudinal research for the long-term protection and efficacy rate of vaccines before a conclusion could be made. Although our findings are not quite relevant to this issue, we have added a brief discussion on this matter:

(p.13) “…whilst others have proposed or launched the “vaccine passport” (a.k.a. “Green Pass”, “COVID Certificate”, “Coronapas”, “VaccineGuard” and “Excelsior Pass”) schemes [47, 48, 49]. Under these schemes, individuals with immunity, whether based on a past recovery from COVID-19 or a completion of two doses of recognised vaccines, may be exempted from free movement restrictions or quarantine policy when traveling, and granted free access to social, sports, and cultural events. By allowing immunised individuals to move freely and progressively return to normal life, vaccine passport schemes may speed up the vaccination uptake in the society. However, there have been doubts on relevant ethical and safety issues… As to safety issues, there have been concerns driven by uncertainties about the length of time that vaccine-built protection could maintain. Evidence is not yet extensive for how far existing vaccines could prevent COVID-19 transmission, and how high the risk of contracting COVID-19 would be after receiving two doses of vaccines is still unknown. Whether it is safe and secure for a group of vaccinated individuals to gather without proper disease preventive measures (such as mask wearing and social distancing) requires further research.”

Round 2

Reviewer 1 Report

All my comments were properly addressed.